# Development and psychometric properties of the Japanese Consumer Assessment of Healthcare Providers and Systems Clinician & Group Survey (CG-CAHPS)

**Takuya Aoki** [1,2]*, **Kuichiro Taguchi**[3], **Eiichi Hama**[3]

**1** Division of Clinical Epidemiology, The Jikei University School of Medicine, Tokyo, Japan, **2** Section of Clinical Epidemiology, Department of Community Medicine, Graduate School of Medicine, Kyoto University, Kyoto, Japan, **3** Institute of Quality and Time of Healthcare, Tokyo, Japan

* taoki@jikei.ac.jp

**Data Availability Statement:** All relevant data are within the manuscript and its Supporting Information files.

## Abstract

The Consumer Assessment of Healthcare Providers and Systems Clinician & Group Survey (CG-CAHPS) is one of the most widely studied and endorsed patient experience measures for ambulatory care. This study aimed to develop a Japanese CG-CAHPS and examine its psychometric properties. We evaluated the structural validity, criterion-related validity, internal consistency reliability, and site-level reliability of the scale. Data were analyzed for 674 outpatients aged 18 years or older in 11 internal medicine clinics. The confirmatory factor analysis supported the scale's structural validity and the same composites (Access, Provider Communication, Care Coordination, and Office Staff) as that of the original CG-CAHPS. All site-level Pearson correlation coefficients between the Japanese CG-CAHPS composites and overall provider rating exceeded the criteria. Results of item-total correlations and Cronbach's alpha indicated adequate internal consistency reliability. We developed the Japanese CG-CAHPS and examined its validity and reliability to measure the quality of ambulatory care based on patient experience. The results of the Japanese CG-CAHPS survey will provide useful information to providers, organizations, and policy makers for achieving a patient-centered healthcare system in Japan.

## Introduction

The quality assessment of patient-centeredness from the patient's perspective is an important aspect of quality of health care [1]. In recent years, better patient experience has been recognized as one of the crucial goals of healthcare alongside population health and per capita cost [2]. Patient experience is integrally tied to the principles and practices of patient- and family-centered care. Embedded within patient experience is a focus on individualized care and tailoring services to meet patients' needs and engage them as partners in their care [3]. Numerous studies have shown that better patient experience is positively associated with patient health outcomes, patient safety, and patient behaviors across a wide range of disease areas and settings [4–7].

**Funding:** This work was supported by JSPS KAKENHI Grant Number JP20K18849. The funder had no role in study design, data collection and analysis, decision to publish, or preparation of the manuscript. The EPARK provided support in the form of salaries for authors [KT and EH], but did not have any additional role in the study design, data collection and analysis, decision to publish, or preparation of the manuscript. The specific roles of these authors are articulated in the 'author contributions' section.

**Competing interests:** I have read the journal's policy and the authors of this manuscript have the following competing interests: KT and EH are an employee of EPARK. This does not alter our adherence to PLOS ONE policies on sharing data and materials.

The Consumer Assessment of Healthcare Providers and Systems (CAHPS®) Clinician & Group Survey (CG-CAHPS) is one of the most widely studied and endorsed patient experience measures for ambulatory care [8]. This standardized scale was developed by the Agency for Healthcare Research and Quality (AHRQ) and its validity and reliability were examined [9–11]. Currently, in the United States, CG-CAHPS results are widely used as quality measures in accountability initiatives and to stimulate, guide, and monitor quality improvement efforts [12]. For example, results from the CG-CAHPS have been reported on the Center for Medicare & Medicaid Services Physician Compare website, and insurers are also increasingly including patient experience data in pay-for-performance programs.

In contrast, in Japan, voluntary activities for the assessment of patient experience have just begun in limited settings, and systematic approaches for quality assessment and improvement based on patient experience measures are still unestablished. Only a few standardized scales, which have been confirmed psychometric properties, are available to assess patient experience in Japan [13–15]. Patient experience scales, which are tailored to different care settings, are needed to help identify aspects of care that can be targeted to improve patient experience. Accordingly, the present study aimed to develop a Japanese version of the CG-CAHPS version 3.0, which asks patients about their experiences with care from an ambulatory care provider over a six-month period, and to examine its structural validity, criterion-related validity, internal consistency reliability, and site-level reliability.

## Materials and methods

### Design, setting, and participants

The data used in this study were collected from a multicenter cross-sectional survey in 11 internal medicine clinics from February to March 2020. The 11 participating clinics, which were cooperating facilities of the Institute of Quality and Time of Healthcare, voluntarily took part in the survey and are in urban areas in the Tokyo Metropolis and Kanagawa Prefecture, with all the clinics being privately owned and managed. In Japan, clinics are generally run by one full-time physician, nurses, and medical assistants, and they provide outpatient services and possibly home care. Outpatient services in the participating clinics were provided by physicians trained in an internal medicine–based residency program in general hospitals.

Independent surveyors distributed a self-administered questionnaire in the waiting room directly to all outpatients aged 18 years or older who visited one of the participating clinics within the survey period using a continuous sampling method. The survey period was two consecutive days. Patients who were assessed by the surveyors as unable to respond to the questionnaire due to severe physical or mental disorders were excluded. According to the sampling criteria of the original CG-CAHPS version 3.0, of the survey respondents, those who responded that they have had at least one visit with the provider in the last six months were included in our analyses.

This study was approved by the Ethics Committee of Kyoto University Graduate School of Medicine (approval number R2331). Informed consent was not obtained because this study is an analysis of existing anonymous data. Data were anonymized before access.

### Measures

**CG-CAHPS.** The original CG-CAHPS version 3.0 (the latest version as of February 2020) is an 18-item tool comprising four composites, a global rating, and five screening items (Q1, Q3, Q5, Q12, and Q15) [8]. The composites are Access (Q2, Q4, and Q6), Provider Communication (Q7, Q8, Q10, and Q11), Care Coordination (Q9, Q13, and Q16), and Office Staff (Q17 and Q18). The global rating is Rating of Provider (Q14).

Permission to translate the CG-CAHPS into Japanese was granted by the AHRQ. Following the translation guidelines for CAHPS® surveys provided by AHRQ [16], translation of the CG-CAHPS into Japanese was performed through the following steps. First, two bilingual translators with experience in translating survey instruments conducted forward translations from English to Japanese. Two forward translations were performed independently. The two translations were then reviewed by a translation reviewer who is a native speaker of Japanese and has experience in translating survey instruments. After reviewing the translation, the reviewer prepared a reconciled version of the translation. A committee consisting of the two translators and the reviewer then discussed and prepared the final version. Revisions were made to the reconciled version necessary for cross-cultural adaptation. The final wording of each survey item and response option was determined by consensus (S1 Appendix).

The CG-CAHPS survey uses multiple response formats: four-point scales (1 = never, 2 = sometimes, 3 = usually, and 4 = always), and a global rating scale (0 = worst to 10 = best). To make the results easier to understand, we converted all scales to scores ranging from 0 to 100 using the following formula:

Converted Score = 100 * (Respondent's selected response value–Minimum response value on the scale) / (Maximum response value—Minimum response value)

In the Japanese version, assuming the convergence in each composite as in the original version, the score for each of the four composites was computed as the mean value for all converted scores in the scale that would fall in the range of 0–100 points, with higher scores indicating better performance.

## Statistical analysis

We validated the Japanese CG-CAHPS through the following steps:

First, we carried out a confirmatory factor analysis based on Pearson correlations using the maximum likelihood estimation to evaluate the structural validity of the Japanese CG-CAHPS composites. In the factor analysis based on Pearson correlations, we hypothesized the same factor structure (four-factor solution) as that of the original CG-CAHPS. In addition, we also conducted a confirmatory factor analysis based on polychoric correlations using the diagonally weighted least squares estimation to address the concern that Pearson correlations may not be appropriate. The factor analysis based on polychoric correlations was limited to three factors: Provider Communication, Care Coordination, and Office Staff. Because many participants responded "Not applicable" to the Access items, it was difficult to estimate a polychoric correlation matrix in the four-factor structure. The appropriateness of the resulting structure was determined by examining if factor loadings were 0.40 or greater [17]. Model fitness was assessed using the comparative fit index (CFI), root mean square error of approximation (RMSEA), and standardized root mean square residual (SRMR). For CFI, a value of > 0.95 indicates goodness of fit. Previous studies suggest that models with RMSEA < 0.06 and SRMR < 0.08 are representative of models with good fit [18].

Second, we used the Japanese CG-CAHPS composite scores and the overall provider rating to examine criterion-related validity. Validity was assessed using Pearson correlation coefficients with each Japanese CG-CAHPS composite to predict the Rating of Provider (0 = Worst to 10 = Best) of the scale at the site-level. A correlation coefficient greater than 0.30 was considered meaningful [19]. Site-level correlations are a more important criterion for measurement than patient-level correlations because the former are benchmarking tools to compare one provider or facility with another. To examine site-level correlations, we used each provider's mean score on CG-CAHPS composites and the Rating of Provider.

Internal consistency reliability was examined by item-total correlations and Cronbach's alpha. We also examined site-level reliability for each score using a linear mixed effect model to adjust for the case-mix variables: age, education, and self-rated health. Site-level reliability was estimated by using the following formula [20]:

Reliability = (Between-site variance) / [Between-site variance + (Within-site variance)/ (Sample size for site)]

For a scale to be considered sufficiently reliable, an item-total correlation of 0.30 and Cronbach's alpha and site-level reliability value of 0.70 are recommended [21].

Finally, descriptive statistics were performed on the Japanese CG-CAHPS scores, including the mean, standard deviation, and observed range. To deal with missing data, in the confirmatory factor analysis based on Pearson correlations, we used the full information maximum likelihood estimation to enable the use of information collected from participants with missing data. In the evaluation of confirmatory factor analysis based on polychoric correlations, criterion-related validity, and reliability, we conducted complete case analyses. All statistical analyses were conducted using R version 3.6.3 (R Foundation for Statistical Computing, Vienna, Austria; www.R-project.org).

## Results

Of the total 818 eligible outpatients, 787 (96.2%) responded to the survey. Of these respondents, we analyzed the data from 674 patients who responded that they have had at least one visit with the provider in the last 6 months. Table 1 shows the participants' characteristics.

Table 2 shows the participants' responses to each item of the Japanese CG-CAHPS. The Top Box score for each item, which is the percentage of participants who provided the most positive responses on that item, ranged from 56.1% to 73.6%. Regarding the mean Top Box score for composites, the highest score was observed for Provider Communication (70.9%), while the lowest score was for Care Coordination (59.6%). The bottom box score, which is the percentage of participants with the least positive responses on the item, ranged from 0.8% to 7.2%.

### Structural validity

Fig 1 shows the path diagrams of the confirmatory factor analysis based on Pearson correlations to assess the structural validity of four-factor structure of the Japanese CG-CAHPS composites. All factor loadings of each item onto each factor were above the 0.40 criteria, ranging from 0.48 to 0.90. The correlation coefficients among factors ranged from 0.51 to 0.89. Part of the model fit indices showed good fit (CFI = 0.940, RMSEA = 0.074, and SRMR = 0.067). The confirmatory factor analysis based on polychoric correlations indicated excellent goodness of fit (CFI = 0.998, RMSEA = 0.054, and SRMR = 0.047).

### Criterion-related validity

Table 3 shows the Pearson correlation coefficients between the Japanese CG-CAHPS composites and the Rating of Provider as an overall provider rating at the site-level. All correlations exceeded the 0.30 criterion. Provider Communication ($r = 0.85$) had the highest correlation with the overall rating.

### Reliability and descriptive statistics

Table 4 indicates the score distribution, internal consistency reliability, and site-level reliability for the Japanese CG-CAHPS. All item-total correlations were above the 0.30 criteria, ranging

**Table 1. Participants' characteristics (N = 674).**

| Characteristic | n (%) |
|---|---|
| Gender | |
| Male | 251 (37.9) |
| Female | 411 (62.0) |
| Data missing | 12 |
| Age (years) | |
| 18–24 | 34 (5.1) |
| 25–34 | 125 (18.8) |
| 35–44 | 114 (17.2) |
| 45–54 | 126 (19.0) |
| 55–64 | 109 (16.4) |
| 65–74 | 86 (13.0) |
| 75 or more | 70 (10.5) |
| Data missing | 10 |
| Education | |
| Less than high school | 34 (5.1) |
| High school | 192 (29.0) |
| Junior college | 136 (20.5) |
| More than or equal to college | 301 (45.4) |
| Data missing | 11 |
| Self-rated Health | |
| Excellent | 19 (2.9) |
| Very good | 77 (11.6) |
| Good | 160 (24.1) |
| Fair | 317 (47.7) |
| Poor | 92 (13.8) |
| Data missing | 9 |
| Duration of relationship with provider | |
| Less than 6 months | 222 (33.3) |
| At least 6 months but less than 1 year | 116 (17.4) |
| At least 1 year but less than 3 years | 147 (22.0) |
| At least 3 years but less than 5 years | 66 (9.9) |
| 5 years or more | 116 (17.4) |
| Data missing | 7 |

from 0.31 to 0.92. For Access, Provider Communication, and Office Staff, the Cronbach's alpha and site-level reliability were above 0.70. However, for Care Coordination, the Cronbach's alpha and site-level reliability did not exceed the 0.70 criterion.

Descriptive statistics showed that the highest scored scale was Provider Communication (mean score = 88.1), and the most poorly scored scale was Care Coordination (mean score = 78.6). The full range of possible scores was observed for all scales.

## Discussion

We developed the Japanese CG-CAHPS in the form of a standardized scale for assessing the quality of ambulatory care from the patient's perspective in Japan. In our multicenter study, the psychometric properties of the Japanese CG-CAHPS, including structural validity, criterion-related validity, internal consistency reliability, and site-level reliability were evaluated. Even in Japan, it is important to have valid and reliable measures for assessing patient experience in

**Table 2. Response to Japanese CG-CAHPS items (N = 674): Number (%).**

| | Never | Sometimes | Usually | Always | Data missing | Not applicable[a] |
|---|---|---|---|---|---|---|
| **Access:** | | | | | | |
| Q2. In the last 6 months, when you contacted this provider's office to get an appointment for care you needed right away, how often did you get an appointment as soon as you needed? | 7 (3.6) | 16 (8.3) | 45 (23.4) | 124 (64.6) | 10 | 472 |
| Q4. In the last 6 months, when you made an appointment for a check-up or routine care with this provider, how often did you get an appointment as soon as you needed? | 11 (4.5) | 27 (11.1) | 43 (17.6) | 163 (66.8) | 12 | 418 |
| Q6. In the last 6 months, when you contacted this provider's office during regular office hours, how often did you get an answer to your medical question that same day? | 5 (3.4) | 16 (10.7) | 32 (21.5) | 96 (64.4) | 0 | 525 |
| **Provider Communication:** | | | | | | |
| Q7. In the last 6 months, how often did this provider explain things in a way that was easy to understand? | 6 (0.9) | 27 (4.1) | 139 (21.3) | 480 (73.6) | 22 | – |
| Q8. In the last 6 months, how often did this provider listen carefully to you? | 8 (1.2) | 25 (3.8) | 148 (22.4) | 480 (72.6) | 13 | – |
| Q10. In the last 6 months, how often did this provider show respect for what you had to say? | 8 (1.2) | 25 (3.8) | 148 (22.7) | 471 (72.2) | 22 | – |
| Q11. In the last 6 months, how often did this provider spend enough time with you? | 8 (1.2) | 41 (6.3) | 178 (27.3) | 426 (65.2) | 21 | – |
| **Care Coordination:** | | | | | | |
| Q9. In the last 6 months, how often did this provider seem to know the important information about your medical history? | 31 (4.7) | 40 (6.1) | 205 (31.4) | 377 (57.7) | 21 | – |
| Q13. In the last 6 months, when this provider ordered a blood test, x-ray, or other test for you, how often did someone from this provider's office follow up to give you those results? | 24 (7.2) | 24 (7.2) | 69 (20.7) | 216 (64.9) | 15 | 326 |
| Q16. In the last 6 months, how often did you and someone from this provider's office talk about all the prescription medicines you were taking? | 34 (5.9) | 73 (12.6) | 147 (25.4) | 324 (56.1) | 20 | 76 |
| **Office Staff:** | | | | | | |
| Q17. In the last 6 months, how often were clerks and receptionists at this provider's office as helpful as you thought they should be? | 9 (1.4) | 49 (7.5) | 214 (32.7) | 383 (58.5) | 19 | – |
| Q18. In the last 6 months, how often did clerks and receptionists at this provider's office treat you with courtesy and respect? | 5 (0.8) | 43 (6.6) | 182 (27.8) | 425 (64.9) | 19 | – |
| | 0–2 | 3–5 | 6–8 | 9–10 | Data missing | Not applicable[a] |
| **Rating of Provider:** | | | | | | |
| Q14. Using any number from 0 to 10, where 0 is the worst provider possible and 10 is the best provider possible, what number would you use to rate this provider? | 3 (0.5) | 33 (5.1) | 204 (31.2) | 414 (63.3) | 20 | – |

[a]The number of participants who skipped the item due to the response to the screening item.

various settings. This scale could be used for quality improvement based on the assessment of patient experience with ambulatory care and for health services research in Japan.

The confirmatory factor analysis supported the scale's structural validity and the same composites (Access, Provider Communication, Care Coordination, and Office Staff) as that of the original CG-CAHPS. Correlation coefficients between all Japanese CG-CAHPS composites and the overall provider rating for assessing criterion-related validity exceeded the meaningful value at the site-level. In internal consistency analyses, only Cronbach's alpha for Care Coordination did not exceed the recommended value; however, all item-total correlations were greater than the cutoff value, which indicated acceptable internal consistency of the scales. The site-level reliability for Care Coordination was 0.68, slightly below the cutoff of 0.70, indicating that it is necessary to have more participants per provider to increase reliability to acceptable levels. The low internal consistency reliability for Care Coordination and the strongest correlation of Provider Communication with overall provider rating were consistent with the results of previous studies [10, 20].

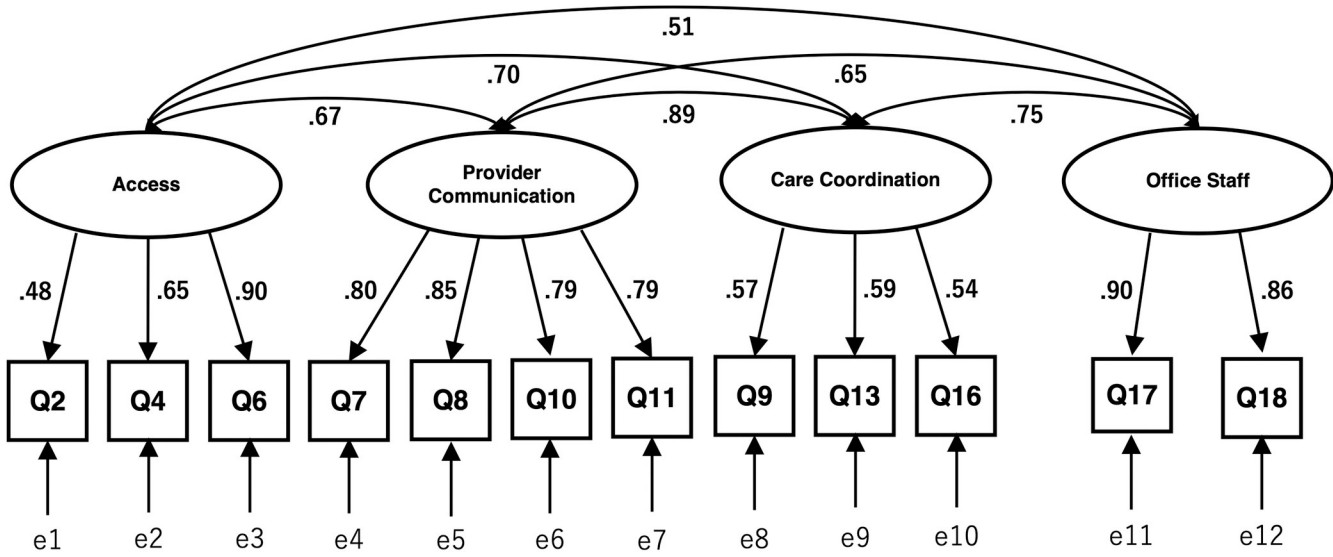

**Fig 1. Factor structure of Japanese CG-CAHPS (confirmatory factor analysis).** Squares are observed variables (items); ellipses are latent variables (factors), values on the single-headed arrows are standardized factor loadings, values on the double-headed arrows are correlation coefficients. CG-CAHPS = Consumer Assessment of Healthcare Providers and Systems Clinician & Group Survey.

**Table 3. Pearson correlation coefficients between Japanese CG-CAHPS composites and overall provider rating.**

| Composites | Site-level correlations |
|---|---|
| Access | 0.59 |
| Provider Communication | 0.85 |
| Care Coordination | 0.79 |
| Office Staff | 0.49 |

**Table 4. Descriptive features, internal consistency reliability, and site-level reliability of Japanese CG-CAHPS (N = 674).**

| | Number of items | Mean | Standard deviation | Observed range | Item-total correlation | Cronbach's alpha | Site-level reliability[a] |
|---|---|---|---|---|---|---|---|
| Composites | | | | | | | |
| Access | 3 | 83.9 | 25.0 | 0.0–100.0 | 0.53–0.92 | 0.81 | 0.77 |
| Provider Communication | 4 | 88.1 | 17.8 | 0.0–100.0 | 0.73–0.78 | 0.88 | 0.83 |
| Care Coordination | 3 | 78.6 | 23.3 | 0.0–100.0 | 0.31–0.44 | 0.58 | 0.68 |
| Office Staff | 2 | 84.2 | 21.0 | 0.0–100.0 | 0.77 | 0.87 | 0.86 |
| Global ratings | | | | | | | |
| Rating of Provider | 1 | 87.9 | 15.6 | 0.0–100.0 | – | – | 0.74 |

[a]Average number of participants per site was 61.

The CG-CAHPS is one of the most widely studied patient experience scales for ambulatory care worldwide. The CG-CAHPS has been translated into many languages in order to be used in other countries so that comparisons of health service quality from the patient perspective can be made. In our study, the recovery rate for the questionnaire administered was very high, suggesting a low risk of selection bias.

However, the present study has several potential limitations. First, in this study, we evaluated the structural validity, criterion-related validity, and internal consistency reliability of the Japanese CG-CAHPS, other psychometric properties, including convergent and discriminant validity, test-retest reliability, and interpretability, have not been assessed [22]. These psychometric properties of the scale need to be evaluated in future studies. Second, our survey setting was restricted to urban areas and may not have sufficiently represented the Japanese national level. Therefore, the study results may have limited generalizability and a survey using the Japanese CG-CAHPS in other suburban and rural areas should be conducted.

## Conclusions

We developed the Japanese CG-CAHPS and examined its validity and reliability to measure the quality of ambulatory care based on patient experience. The results of the Japanese CG-CAHPS survey will provide useful information to providers, organizations, and policy makers for achieving a patient-centered healthcare system in Japan.

## Supporting information

**S1 Appendix.**
(PDF)

**S1 File.**
(XLSX)

## Acknowledgments

We would like to thank Yukiko Matsuo (Ohana clinic Tokyo), So Ishii (Kudan-shita Eki-mae Coco Clinic), Tsuyoshi Sugimoto (Seint Clinic Ikebukuro Ekimae), Toshiyuki Kaneko (Tokyo skytree medical clinic) for their support during the study period.

## Author Contributions

**Conceptualization:** Takuya Aoki, Kuichiro Taguchi, Eiichi Hama.

**Data curation:** Takuya Aoki, Kuichiro Taguchi, Eiichi Hama.

**Formal analysis:** Takuya Aoki.

**Funding acquisition:** Takuya Aoki.

**Investigation:** Takuya Aoki, Kuichiro Taguchi, Eiichi Hama.

**Methodology:** Takuya Aoki.

**Project administration:** Takuya Aoki, Kuichiro Taguchi, Eiichi Hama.

**Resources:** Takuya Aoki, Kuichiro Taguchi, Eiichi Hama.

**Writing – original draft:** Takuya Aoki.

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
