## [Decision Letter · Decision Letter 0]

22 Feb 2021

PONE-D-20-24361

Development and psychometric properties of the Japanese Consumer Assessment of Healthcare Providers and Systems Clinician & Group Survey (CG-CAHPS)

PLOS ONE

Dear Dr. Aoki,

Thank you for submitting your manuscript to PLOS ONE. After careful consideration, we feel that it has merit but does not fully meet PLOS ONE’s publication criteria as it currently stands. Therefore, we invite you to submit a revised version of the manuscript that addresses the points raised during the review process.

We look forward to receiving your revised manuscript.

Kind regards,

Stefan Hoefer

Academic Editor

PLOS ONE

Journal Requirements:

2. In your methods section you indicate that "informed consent was not obtained because this is an analysis of existing anonymous data." However, you also mention that you excluded from this survey, patients who visited the participating clinics for the first time. There is significant discord between your two statements. Please clarify whether you collected primary data or secondary data? If primary data collection was done, please justify why informed consent was not obtained. If this is secondary data analysis, please explain if data were de-identified/ anonymised before access.

"I have read the journal's policy and the authors of this manuscript have the following competing interests: Mr. Kuichiro Taguchi and Mr. Eiichi Hama are an employee of EPARK. "

We note that one or more of the authors are employed by a commercial company: EPARK;  Institute of Quality and Time of Healthcare

(2) Please also provide an updated Competing Interests Statement declaring this commercial affiliation along with any other relevant declarations relating to employment, consultancy, patents, products in development, or marketed products, etc.  

Reviewers' comments:

Reviewer's Responses to Questions

**Comments to the Author**

1. Is the manuscript technically sound, and do the data support the conclusions?

Reviewer #1: Yes

Reviewer #2: Yes

2. Has the statistical analysis been performed appropriately and rigorously? 

Reviewer #1: Yes

Reviewer #2: Yes

3. Have the authors made all data underlying the findings in their manuscript fully available?

Reviewer #1: Yes

Reviewer #2: Yes

4. Is the manuscript presented in an intelligible fashion and written in standard English?

Reviewer #1: Yes

Reviewer #2: Yes

5. Review Comments to the Author

Reviewer #1: This is a well done study and nicely described paper. However, I have several suggestions to improve the work.

1) It is good to see that the authors acknowledge the provider-level measurement when computing correlations among the CAHPS measures. But they also need to report physician-level reliability estimates. Categorical factor analysis (polychoric correlations) should be conducted. The text about coefficient alpha on lines 213-216 is incorrect and should be deleted. Yes, alpha will increase with the number of items as long as they are positively correlated with one another, but the fact still remains that a shorter test with low reliability has low reliability. Because of what is noted above about the use of CAHPS, the important reliability estimate is the one directed at the unit of application of the measure (provider-level).

2) Most of their results jive with what has been found in CAHPS before (e.g., low internal consistency reliability for care coordination, strongest correlations of communication with overall provider rating). Referring back to other literature that has shown this would be helpful.

3) How as the survey distributed and where (line 78)?

4) What is mean by "within two days of the survey period" (line 79)?

5) How was it determined that potential respondents had a "severe physical or mental" (line 81) disorder?

6) When the authors refer to the Japanese version of the C-G CAHPS survey (e.g., line 68) they should make it clear that they are referring to the CAHPS Clinician & Group Survey 3.0 that asks patients about their experiences with care from an ambulatory care provider over a six-month period. (Don't wait until line 89). Relatedly, reference 10 is for another C-G survey, not the one used in this study. Some references that are relevant are the following. These references refer to the patient-centered medical home survey but the C-G items are embedded within it:

Scholle, S. H., et al. (2012). Development of and field-test results for the CAHPS PCMH survey. Medical Care, 50, S2-10.

Hays, R. D., et al. (2014). Evaluating the psychometric properties of the CAHPS patient-centered medical home survey. Clinical Therapeutics, 36 (5), 689-696. PMCID: PMC4087122.

And on lines 81-82 when it is mentioned that those visiting the clinic for the first time were excluded, say why. Given that the survey asks about care during the last 6 months, why did were those who had a duration of less than 6 months (33% of the sample, Table 1) included in the study?

7) The fit cutoffs indicated are dated. See somewhat more recent thresholds such as: Reeve, B. B. et al.(2007). Psychometric evaluation and calibration of health-related quality of life item banks: Plans for the Patient-Reported Outcome Measurement Information System (PROMIS). Medical Care, 45, S22-31.

Also:

* When the CAHPS abbreviation is used the first time, the registered copyright symbol should follow it.

"consistently associated with" (line 52)

- In what direction?

* What does "confirmed its validity and reliability" (line 57)and a "valid and reliable scale" (line 233) mean? Reliability and validity are not dichotomous attributes.

* "Physician Compare website" (line 60) -> Center for Medicare & Medicaid Services Physician

Compare website.

* The never to always response scale is a polytomous, but it is not a "Likert scale": http://www.john-uebersax.com/stat/likert.htm

* The 0-100 possible range score is not "normalized" in the psychometric or statistical sense so I would not describe it is such (line 109, 113).

Reviewer #2: Thank you for the opportunity to review your manuscript. I found it to be generally well-written. I would appreciate some clarification on a small number of issues, as below:

Material and methods

Page 10 Line 74

Please explain how you selected the 11 internal medicine clinics. Are there any characteristics for the physicians among the selected 11 clinics? In Japan, physicians’ practice depends on their physician’s career.

Measures

Page 11 Line 98

For the cross-cultural adaptation of the development of questionnaires, conducting the back-translation process is mostly included as follows.

https://journals.lww.com/spinejournal/Fulltext/2000/12150/Guidelines_for_the_Process_of_Cross_Cultural.14.aspx

Did you do back-translation or remove the backward translation process?

If you removed the backward translation process, please explain.

Discussion

Page 19 Line 214-215

Are there any other reasons why Cronbach’s alpha for Coordination did not exceed the recommended value? For example, the each items’ difference or characteristics among the scale.

6. PLOS authors have the option to publish the peer review history of their article (what does this mean?). If published, this will include your full peer review and any attached files.

Reviewer #1: **Yes: **Ron D. Hays

Reviewer #2: No

---

## [Author Response · Author response to Decision Letter 0]

6 Mar 2021

March 5, 2021

Dear Editor;

Re: Manuscript reference No. PONE-D-20-24361

Please find attached a revised version of our manuscript “Development and psychometric properties of the Japanese Consumer Assessment of Healthcare Providers and Systems Clinician & Group Survey (CG-CAHPS)”, which we would like to resubmit for publication as an original research article in PLOS ONE. 

Your comments and those of the reviewers were highly insightful and enabled us to greatly improve the quality of our manuscript. In the following pages are our point-by-point responses to each of the comments of the reviewers.

We hope that the revisions in the manuscript and our accompanying responses will be sufficient to make our manuscript suitable for publication in PLOS ONE.

We shall look forward to hearing from you at your earliest convenience.

Yours sincerely,

Takuya Aoki

Division of Clinical Epidemiology, The Jikei University School of Medicine, 3-25-8 Nishishimbashi, Minato-ku, Tokyo, 105-8461, Japan.

Tel: +81-3-3433-1111; E-mail: taoki@jikei.ac.jp

Responses to the comments of Editor

In your methods section you indicate that "informed consent was not obtained because this is an analysis of existing anonymous data." However, you also mention that you excluded from this survey, patients who visited the participating clinics for the first time. There is significant discord between your two statements. Please clarify whether you collected primary data or secondary data? If primary data collection was done, please justify why informed consent was not obtained. If this is secondary data analysis, please explain if data were de-identified/ anonymised before access.

Response: Our study was secondary data analysis. We have improved the description in the Design, setting, and participants section as follows. 

According to the sampling criteria of the original CG-CAHPS version 3.0, of the survey respondents, those who responded that they have had at least one visit with the provider in the last six months were included in our analyses. 

This study was approved by the Ethics Committee of Kyoto University Graduate School of Medicine (approval number R2331). Informed consent was not obtained because this study is an analysis of existing anonymous data.

Thank you for stating the following in the Competing Interests section:

"I have read the journal's policy and the authors of this manuscript have the following competing interests: Mr. Kuichiro Taguchi and Mr. Eiichi Hama are an employee of EPARK. "

We note that one or more of the authors are employed by a commercial company: EPARK; Institute of Quality and Time of Healthcare

Response: Accordingly, we have updated the Funding Statement and Competing Interests Statement and included them in our cover letter. 

Responses to the comments of Reviewer #1

We wish to express our appreciation to you for your insightful comments on our paper. The comments have helped us significantly improve the paper. 

1) It is good to see that the authors acknowledge the provider-level measurement when computing correlations among the CAHPS measures. But they also need to report physician-level reliability estimates. Categorical factor analysis (polychoric correlations) should be conducted. The text about coefficient alpha on lines 213-216 is incorrect and should be deleted. Yes, alpha will increase with the number of items as long as they are positively correlated with one another, but the fact still remains that a shorter test with low reliability has low reliability. Because of what is noted above about the use of CAHPS, the important reliability estimate is the one directed 

at the unit of application of the measure (provider-level).

Response: Thank you for your comments. Accordingly, we have added the descriptions of site-level reliability and factor analysis based on polychoric correlations to the Statistical analysis and Results sections as follows.

Statistical analysis

First, we carried out a confirmatory factor analysis based on Pearson correlations using the maximum likelihood estimation to evaluate the structural validity of the Japanese CG-CAHPS composites. In the factor analysis based on Pearson correlations, we hypothesized the same factor structure (four-factor solution) as that of the original CG-CAHPS. In addition, we also conducted a confirmatory factor analysis based on polychoric correlations using the diagonally weighted least squares estimation to address the concern that Pearson correlations may not be appropriate. The factor analysis based on polychoric correlations was limited to three factors: Provider Communication, Care Coordination, and Office Staff. Because many participants responded "Not applicable" to the Access items, it was difficult to estimate a polychoric correlation matrix in the four-factor structure.

Internal consistency reliability was examined by item-total correlations and Cronbach’s alpha. We also examined site-level reliability for each score using a linear mixed effect model to adjust for the case-mix variables: age, education, and self-rated health. Site-level reliability was estimated by using the following formula [20]:

Reliability = (Between-site variance) / [Between-site variance + (Within-site variance)/(Sample size for site)]

For a scale to be considered sufficiently reliable, an item-total correlation of 0.30 and Cronbach’s alpha and site-level reliability value of 0.70 are recommended [21].

Results

Fig 1 shows the path diagrams of the confirmatory factor analysis based on Pearson correlations to assess the structural validity of four-factor structure of the Japanese CG-CAHPS composites. All factor loadings of each item onto each factor were above the 0.40 criteria, ranging from 0.48 to 0.90. The correlation coefficients among factors ranged from 0.51 to 0.89. Part of the model fit indices showed good fit (CFI = 0.940, RMSEA = 0.074, and SRMR = 0.067). The confirmatory factor analysis based on polychoric correlations indicated excellent goodness of fit (CFI = 0.998, RMSEA = 0.054, and SRMR = 0.047).

Table 4 indicates the score distribution, internal consistency reliability, and site-level reliability for the Japanese CG-CAHPS. All item-total correlations were above the 0.30 criteria, ranging from 0.31 to 0.92. For Access, Provider Communication, and Office Staff, the Cronbach’s alpha and site-level reliability were above 0.70. However, for Care Coordination, the Cronbach’s alpha and site-level reliability did not exceed the 0.70 criterion.

In addition, we have removed the text about Cronbach alpha you pointed from the Discussion section.

2) Most of their results jive with what has been found in CAHPS before (e.g., low internal consistency reliability for care coordination, strongest correlations of communication with overall provider rating). Referring back to other literature that has shown this would be helpful.

Response: Thank you for your comments. Accordingly, we have improved the description in the Discussion section as follows.

The low internal consistency reliability for Care Coordination and the strongest correlation of Provider Communication with overall provider rating were consistent with the results of previous studies [10,20].

3) How as the survey distributed and where (line 78)?

Response: We agree that this point requires clarification. Accordingly, we have improved the descriptions of the Design, setting, and participants as follows. 

Independent surveyors distributed a self-administered questionnaire in the waiting room directly to all outpatients aged 18 years or older who visited one of the participating clinics within the survey period using a continuous sampling method.

4) What is mean by "within two days of the survey period" (line 79)?

Response: We have added the sentence as follows.

The survey period was two consecutive days.

5) How was it determined that potential respondents had a "severe physical or mental" (line 81) disorder?

Response: We agree that this point requires clarification. Accordingly, we have improved the descriptions to the Design, setting, and participants as follows.

Patients who were assessed by the surveyors as unable to respond to the questionnaire due to severe physical or mental disorders were excluded.

6) When the authors refer to the Japanese version of the C-G CAHPS survey (e.g., line 68) they should make it clear that they are referring to the CAHPS Clinician & Group Survey 3.0 that asks patients about their experiences with care from an ambulatory care provider over a six-month period. (Don't wait until line 89). Relatedly, reference 10 is for another C-G survey, not the one used in this study. Some references that are relevant are the following. These references refer to the patient-centered medical home survey but the C-G items are embedded within it:

Scholle, S. H., et al. (2012). Development of and field-test results for the CAHPS PCMH survey. Medical Care, 50, S2-10.

Hays, R. D., et al. (2014). Evaluating the psychometric properties of the CAHPS patient-centered medical home survey. Clinical Therapeutics, 36 (5), 689-696. PMCID: PMC4087122.

Response: Thank you for your comments. Accordingly, we have replaced the reference to them you suggested and improved the description of Introduction section as follows.

Accordingly, the present study aimed to develop a Japanese version of the CG-CAHPS version 3.0, which asks patients about their experiences with care from an ambulatory care provider over a six-month period, and to examine its structural validity, criterion-related validity, internal consistency reliability, and provider-level reliability.

And on lines 81-82 when it is mentioned that those visiting the clinic for the first time were excluded, say why. Given that the survey asks about care during the last 6 months, why did were those who had a duration of less than 6 months (33% of the sample, Table 1) included in the study?

Response: Accordingly, we have added this point to the Methods section as follows. Incidentally, duration of relationship with provider is independent of sampling method in the CG-CAHPS survey.

According to the sampling criteria of the original CG-CAHPS version 3.0, of the survey respondents, those who responded that they have had at least one visit with the provider in the last six months were included in our analyses. 

7) The fit cutoffs indicated are dated. See somewhat more recent thresholds such as: Reeve, B. B. et al.(2007). Psychometric evaluation and calibration of health-related quality of life item banks: Plans for the Patient-Reported Outcome Measurement Information System (PROMIS). Medical Care, 45, S22-31.

Response: Thank you for your comments. Accordingly, we have improved the description about fit cutoffs in the Statistical analysis section as follows.

The appropriateness of the resulting structure was determined by examining if factor loadings were 0.40 or greater [17]. Model fitness was assessed using the comparative fit index (CFI), root mean square error of approximation (RMSEA), and standardized root mean square residual (SRMR). For CFI, a value of > 0.95 indicates goodness of fit. Previous studies suggest that models with RMSEA < 0.06 and SRMR < 0.08 are representative of models with good fit [18].

Also:

* When the CAHPS abbreviation is used the first time, the registered copyright symbol should follow it.

Response: Thank you for your comments. We have improved this point.

"consistently associated with" (line 52)

- In what direction?

Response: We have improved this text as follows. 

Numerous studies have shown that better patient experience is positively associated with patient health outcomes, patient safety, and patient behaviors across a wide range of disease areas and settings [4-7].

* What does "confirmed its validity and reliability" (line 57)and a "valid and reliable scale" (line 233) mean? Reliability and validity are not dichotomous attributes.

Response: Accordingly, we have improved these sentences as follows.

This standardized scale was developed by the Agency for Healthcare Research and Quality (AHRQ) and its validity and reliability were examined [9-11].

We developed the Japanese CG-CAHPS and examined its validity and reliability to measure the quality of ambulatory care based on patient experience.

* "Physician Compare website" (line 60) -> Center for Medicare & Medicaid Services Physician Compare website.

Response: Thank you for your comments. We have improved this point.

* The never to always response scale is a polytomous, but it is not a "Likert scale": http://www.john-uebersax.com/stat/likert.htm

Response: Thank you for your comments. We have removed the words “Likert scale”.

* The 0-100 possible range score is not "normalized" in the psychometric or statistical sense so I would not describe it is such (line 109, 113).

Response: Accordingly, we have removed the word “normalized”.

Responses to the comments of Reviewer #2

We wish to express our appreciation to you for your insightful comments on our paper. The comments have helped us significantly improve the paper. 

Material and methods

Page 10 Line 74

Please explain how you selected the 11 internal medicine clinics. Are there any characteristics for the physicians among the selected 11 clinics? In Japan, physicians’ practice depends on their physician’s career.

Response: We agree that this point requires clarification. Accordingly, we have improved the descriptions of the Design, setting, and participants as follows. 

The 11 participating clinics, which were cooperating facilities of the Institute of Quality and Time of Healthcare, voluntarily took part in the survey and are in urban areas in the Tokyo Metropolis and Kanagawa Prefecture, with all the clinics being privately owned and managed.

Outpatient services in the participating clinics were provided by physicians trained in an internal medicine–based residency program in general hospitals.

Measures

Page 11 Line 98

For the cross-cultural adaptation of the development of questionnaires, conducting the back-translation process is mostly included as follows.

https://journals.lww.com/spinejournal/Fulltext/2000/12150/Guidelines_for_the_Process_of_Cross_Cultural.14.aspx

Did you do back-translation or remove the backward translation process?

If you removed the backward translation process, please explain.

Response: Thank you for your comments. For translating the CAHPS into other languages, researchers must follow the guidelines provided by the AHRQ. This translation approach involves using two translators to each produce a forward translation and then having the two forward translations reviewed against each other and compared to the original English survey. The advantages of this approach—compared to a simple translation/back-translation approach, for example—include the following: 

• Increased ability to identify and resolve translation errors (i.e., errors in syntax, grammar, or meaning) 

• Increased ability to identify issues related to variations in terms or expressions used by subgroups of people in the target language 

• Increased ability to produce a translation that uses language that is more easily understood by a wide variety of speakers of the target language 

• Increased ability to identify and resolve problems with the readability level of the translation

Agency for Healthcare Research and Quality. Translating CAHPS Surveys. https://www.ahrq.gov/cahps/surveys-guidance/helpful-resources/resources/cahpsGuidelines_Translation.html

Discussion

Page 19 Line 214-215

Are there any other reasons why Cronbach’s alpha for Coordination did not exceed the recommended value? For example, the each items’ difference or characteristics among the scale.

Response: Thank you for your comments. We have added the analysis of site-level reliability as the important reliability estimate and improved the Discussion section as follows.

In internal consistency analyses, only Cronbach’s alpha for Care Coordination did not exceed the recommended value; however, all item-total correlations were greater than the cutoff value, which indicated acceptable internal consistency of the scales. The site-level reliability for Care Coordination was 0.68, slightly below the cutoff of 0.70, indicating that it is necessary to have more participants per provider to increase reliability to acceptable levels.

---

## [Decision Letter · Decision Letter 1]

15 Apr 2021

Development and psychometric properties of the Japanese Consumer Assessment of Healthcare Providers and Systems Clinician & Group Survey (CG-CAHPS)

PONE-D-20-24361R1

Dear Dr. Aoki,

We’re pleased to inform you that your manuscript has been judged scientifically suitable for publication and will be formally accepted for publication once it meets all outstanding technical requirements.

Kind regards,

Stefan Hoefer

Academic Editor

PLOS ONE

Additional Editor Comments (optional):

Reviewers' comments:

Reviewer's Responses to Questions

**Comments to the Author**

1. If the authors have adequately addressed your comments raised in a previous round of review and you feel that this manuscript is now acceptable for publication, you may indicate that here to bypass the “Comments to the Author” section, enter your conflict of interest statement in the “Confidential to Editor” section, and submit your "Accept" recommendation.

Reviewer #2: All comments have been addressed

2. Is the manuscript technically sound, and do the data support the conclusions?

Reviewer #2: Yes

3. Has the statistical analysis been performed appropriately and rigorously? 

Reviewer #2: Yes

4. Have the authors made all data underlying the findings in their manuscript fully available?

Reviewer #2: Yes

5. Is the manuscript presented in an intelligible fashion and written in standard English?

Reviewer #2: Yes

6. Review Comments to the Author

Reviewer #2: (No Response)

7. PLOS authors have the option to publish the peer review history of their article (what does this mean?). If published, this will include your full peer review and any attached files.

Reviewer #2: No

---

## [Editor Report · Acceptance letter]

19 Apr 2021

PONE-D-20-24361R1 

Development and psychometric properties of the Japanese Consumer Assessment of Healthcare Providers and Systems Clinician & Group Survey (CG-CAHPS) 

Dear Dr. Aoki:

I'm pleased to inform you that your manuscript has been deemed suitable for publication in PLOS ONE. Congratulations! Your manuscript is now with our production department. 

Kind regards, 

on behalf of

Dr. Stefan Hoefer 

Academic Editor

PLOS ONE